# Focusing on Ischemic Reperfusion Injury in the New Era of Dynamic Machine Perfusion in Liver Transplantation

**DOI:** 10.3390/ijms25021117

**Published:** 2024-01-17

**Authors:** Gabriela Chullo, Arnau Panisello-Rosello, Noel Marquez, Jordi Colmenero, Merce Brunet, Miguel Pera, Joan Rosello-Catafau, Ramon Bataller, Juan Carlos García-Valdecasas, Yiliam Fundora

**Affiliations:** 1Service of Digestive, Hepato-Pancreatico-Biliary and Liver Transplant Surgery, Institut Clínic de Malalties Digestives i Metabòliques (ICMDM), Hospital Clinic of Barcelona, 08036 Barcelona, Spain; chullo@clinic.cat (G.C.); pera@clinic.cat (M.P.); jcvalde@clinic.cat (J.C.G.-V.); 2Institut d’Investigacions Biomediques August Pi i Sunyer (IDIBAPS), University of Barcelona, 08036 Barcelona, Spain; jcolme@clinic.cat (J.C.); mbrunet@clinic.cat (M.B.); bataller@clinic.cat (R.B.); 3Hepato-Pancreatico-Biliary and Liver Transplant Surgery, Institut Clínic de Malalties Digestives i Metabòliques (ICMDM), Hospital Clinic of Barcelona, 08036 Barcelona, Spain; namarquez@clinic.cat; 4Liver Transplant Unit, Service of Hepatology, Institut Clínic de Malalties Digestives i Metabòliques (ICMDM), Hospital Clinic of Barcelona, 08036 Barcelona, Spain; 5Centro de Investigación Biomédica en Red de Enfermedades hepaticas y digestives (CIBERehd), University of Barcelona, 08036 Barcelona, Spain; 6Pharmacology and Toxicology Laboratory, Biochemistry and Molecular Genetics Department, Biomedical Diagnostic Center, Hospital Clinic of Barcelona, 08036 Barcelona, Spain; 7Experimental Pathology, Institut d’Investigacions Biomèdiques de Barcelona-Consejo Superior de Investigaciones Científicas (IBB-CSIC), 08036 Barcelona, Spain; joan.rosello.catafau@gmail.com

**Keywords:** ischemic reperfusion injury, machine perfusion, liver transplantation, marginal graft, therapeutic strategies

## Abstract

Liver transplantation is the most effective treatment for end-stage liver disease. Transplant indications have been progressively increasing, with a huge discrepancy between the supply and demand of optimal organs. In this context, the use of extended criteria donor grafts has gained importance, even though these grafts are more susceptible to ischemic reperfusion injury (IRI). Hepatic IRI is an inherent and inevitable consequence of all liver transplants; it involves ischemia-mediated cellular damage exacerbated upon reperfusion and its severity directly affects graft function and post-transplant complications. Strategies for organ preservation have been constantly improving since they first emerged. The current gold standard for preservation is perfusion solutions and static cold storage. However, novel approaches that allow extended preservation times, organ evaluation, and their treatment, which could increase the number of viable organs for transplantation, are currently under investigation. This review discusses the mechanisms associated with IRI, describes existing strategies for liver preservation, and emphasizes novel developments and challenges for effective organ preservation and optimization.

## 1. Introduction

Liver transplantation stands as a life-saving therapeutic avenue for individuals afflicted with end-stage liver disease, acute liver failure, and a subset of patients grappling with primary and secondary hepatic malignancies [1]. The expanding spectrum of indications for liver transplantation has engendered a surge in the number of patients awaiting this procedure. Consequently, there exists an imperative to augment the pool of potential liver donors. This situation has necessitated the broadening of acceptance criteria for liver grafts, encompassing the incorporation of increasingly extended criteria donor grafts, colloquially referred to as marginal livers. These marginal livers comprise donations from older individuals, those with hepatic steatosis, grafts subjected to prolonged cold ischemia times (CITs), and donations procured after circulatory death (DCD) [2]. The liver is an organ with many synthetic and secretion functions that receives a dual blood supply, which makes it more susceptible to ischemia and hypoxia [3]. This type of damage can be caused by liver surgery, liver transplantation, and systemic shock. In the case of transplantation, allograft ischemia–reperfusion injury is an inevitable process that contributes to organ shortage and may lead to primary nonfunction or early allograft dysfunction, making livers from marginal donors more susceptible [4,5]. 

IRI manifests as a biphasic process, with the initial phase (ischemia) commencing when hepatocytes face oxygen deprivation following an abrupt cessation of blood flow. Cellular hypoxia disrupts electron flow within the respiratory chain, leading to the depletion of adenosine triphosphate (ATP) and the accumulation of detrimental metabolites and ions, including Na^+^, K^+^, and Ca^++^, due to the failure of ATP-dependent ion channels. This sequence eventually triggers cellular injury, particularly marked by cellular edema during prolonged ischemia, which activates proteases and initiates the apoptotic cascade [6].

A pivotal facet during hypoxia is the accrual of Krebs cycle metabolites, notably succinate, which precipitates the immediate release of reactive oxygen species (ROS) upon the reintroduction of oxygen [7,8].

The subsequent phase (reperfusion) constitutes an exacerbation of the initial insult and is initiated when liver blood flow, oxygen levels, and pH are restored. This restoration unleashes a cascade of injurious components, including proinflammatory cytokines, ROS, and later, aseptic inflammation, which amplifies the damage initiated during ischemia due to metabolic disturbances and the induction of a proinflammatory immune response, expediting cellular degeneration and hastening the decline in liver function [9]. Regrettably, this scenario is further exacerbated when employing marginal grafts sourced from extended criteria donors. Mitochondria are critical players in energy and glucose metabolism, as well as in the regulation of several signaling pathways [10]. Tissue ischemia, resulting from interruption of oxygen supply, severely compromises mitochondrial energy production, and depending on the duration of the ischemic damage, organ dysfunction or failure can be set [11].

Mitochondria have been demonstrated to play a pivotal role in the initiation and modulation of IRI. However, the precise underlying mechanisms of this remain incompletely elucidated, and effective therapeutic interventions are currently lacking.

Given the absence of established treatments for IRI prevention, the primary strategy for mitigating IRI revolves around minimizing the cold preservation time and optimizing graft rewarming. While ex vivo machine perfusion (MP) preservation presents a promising avenue and is gaining recognition, static cold storage (SCS) remains the gold standard for liver preservation. Ex vivo MP holds the capacity to sustain microcirculation, supply requisite substrates and oxygen for cellular metabolism, facilitate the administration of pharmacological or cytoprotective agents directly to the liver, and enable real-time monitoring of their therapeutic efficacy and organ viability during perfusion [12].

A more comprehensive explanation centered on the mechanical and molecular underpinnings of liver IRI would facilitate the identification of novel biomarkers and the discovery of new therapeutic targets. This, in turn, could lead to the establishment of more precise selection criteria and the formulation of innovative strategies aimed at enhancing the surgical outcomes of liver transplantation.

The aim of this review is to provide an overview of the underlying mechanisms of liver IRI and address different protective strategies used during ex vivo liver machine preservation, focusing on current clinical applications and future perspectives in liver transplantation.

## 2. Graft Vulnerability before Preservation: Pre Preservation Injury

### 2.1. Liver Steatosis

Hepatic steatosis is closely linked to conditions such as obesity, alcohol abuse, diabetes, and metabolic disorders. Currently, steatotic livers constitute the majority of extended criteria donor (ECD) grafts, contributing to diminished donor quality and consequently reduced organ utilization [13].

Hepatic steatosis induces alterations in liver grafts at the mitochondrial, Kupffer cell, and sinusoid levels, exacerbating cellular damage during cold ischemia and intensifying ischemia–reperfusion injury [14,15]. The critical challenge lies in discerning the type and extent of macrosteatosis when evaluating the suitability of liver grafts for transplantation [16]. Mild steatosis (<30%) exerts minimal influence on post-transplant outcomes, provided that the cold ischemia time remains brief [15]. Conversely, elevated levels of steatosis are associated with liver failure, graft dysfunction, renal impairment, and biliary complications [17,18]. An analysis of the Scientific Registry of Transplant Recipients revealed that liver allografts with macrovesicular steatosis exceeding 30% independently predicted reduced one-year graft survival [19].

### 2.2. Aged Liver

Liver transplantation has experienced a notable impact due to the utilization of older donor organs, which currently constitute a substantial and increasing segment of the donor pool. Aged livers exhibit significant deregulation of the sinusoidal architecture, thereby enhancing the susceptibility of their sinusoids to acute injuries [20]. Moreover, the synthetic, excretory, and metabolic functions of the liver are susceptible to age-related alterations [21].

Remarkably, outcomes following liver transplantation using organs from older donors have been reported to be on par with those from younger donors, as evidenced by several published studies. These studies demonstrate favorable outcomes with livers procured from donors aged 60, 70, or even 80 years [17].

It is important to note that older donors possess a reduced physiologic reserve and often present with comorbidities that can potentially impact organ procurement. Characteristic features of older liver allografts encompass reduced size, steatosis, capsular fibrosis, and arterial atherosclerosis [21].

### 2.3. Injury Associated with Cardiopulmonary Arrest or Hypotension before Donation

Following cardiac arrest, the cessation of blood flow initiates a progressive depletion of the energy reserves within the donor organ. This depletion leads to the failure of transmembrane ionic pumps and the subsequent development of edema. Furthermore, the absence of perfusion in the affected tissue can result in erythrocyte aggregation and microthrombus formation, which, when combined with the flushing of preservation solution, may further disrupt microcirculation during ischemic transport, potentially compromising tissue viability [22].

Within the literature, research data highlight the divergent impacts of ischemia on liver IRI, contingent on the duration of the ischemic event. It has been documented that a 60 min episode of warm ischemia leads to reversible cellular injury, as liver oxygen consumption returns to control levels upon reperfusion with oxygen. However, reperfusion following longer periods of warm ischemia (120–180 min) results in irreversible cellular damage. These findings align with previous research on hepatic IRI, revealing that hepatocytes reach a point of no return following 90 min of ischemia [23].

Experimental models of partial liver IRI, characterized by ischemic periods of 30–45 min, have indicated a limited role for inducible nitric oxide synthase in injury. However, with ischemia exceeding 60 min, alterations in the expression of inducible nitric oxide synthase become consequential in liver IRI injury [24].

Donors who experience delayed cardiac resuscitation are prone to severe IRI, which can exert a detrimental impact on the initial graft function, particularly when involving extended criteria donor organs.

### 2.4. Injury during Organ Procurement

Injury during the process of organ procurement can arise because of intraoperative hypotension and uncontrolled transient ischemia occurring during donor hepatectomy. This form of injury is predominantly ascribed to the hemodynamic instability of the donor during the procurement procedure, rather than any technical mishaps [5]. Studies have illustrated the adverse correlations between the duration of donor hepatectomy and early allograft dysfunction, ischemic cholangiopathy, as well as graft and patient survival following liver transplantation [25].

### 2.5. Warm Ischemia Injury in Donation after Cardiac Death

In contrast to organ donation after brain death (DBD), the liver in DCD incurs damage prior to preservation, occurring following the withdrawal of life support. The origin of this damage can be traced to warm ischemia initiated by cardiac arrest prior to organ procurement. Extubation marks the commencement of warm ischemia, characterized by inadequate perfusion and tissue oxygenation. There is substantiated evidence indicating that a shorter interval between donor extubation and cardiac arrest (referred to as time after extubation) is associated with a reduced risk of graft failure. Should the cumulative duration of warm ischemia exceed 30 min, numerous transplant centers opt not to accept the liver for transplantation [26,27].

Uncontrolled DCD procedures involve extended periods of warm ischemia, intentionally amplifying cellular damage through inflammatory responses, oxidative stress, and apoptotic processes [28].

## 3. Graft Injury Prevention during Organ Preservation

Liver preservation stands as a critical determinant in the realm of liver transplantation, exerting a profound influence on post-transplant outcomes [29]. Preservation injury encompasses a series of events that unfold before, during, and after liver preservation. In brain-dead donors, this injury stems from four sources: (1) pre-storage injury, (2) cold injury, (3) rewarming injury, and (4) reperfusion injury (Figure 1). Allografts procured after circulatory death are particularly susceptible to heat-related injury.

Organ preservation, defined as “the process of supplying organs for transplantation” [30], plays a pivotal role in ensuring the successful transport and distribution of grafts from distant locations to hospitals where liver transplantation procedures are performed. Over time, organ preservation techniques have evolved significantly. The conventional method, SCS, has served as the gold standard [30]. SCS is often favored for its simplicity, efficient preservation, and ease of organ transfer between donor organ recovery centers and recipient centers. Nevertheless, a major drawback of SCS lies in its limited preservation duration, which inevitably leads to IRI and associated complications, often resulting in graft dysfunction.

### 3.1. Static Cold Graft Storage Injury (SCSI)

Static cold preservation serves as the prevailing method for organ preservation, commencing with the induction of hypothermia. This hypothermic state is initiated during organ storage by perfusing cryopreservation fluid through the artery and portal vein. Therefore, the liver’s temperature rapidly plummets to approximately 16 °C, facilitated by the simultaneous application of ice to the organ’s surface and the abdominal cavity. During cold storage, the intraparenchymal temperature stabilizes at 0–4 °C over nearly one hour [31,32]. The hepatic core temperature reaches a state of equilibrium near 0 °C during static cold storage [32].

Crucially, low temperature stands as the linchpin in maintaining anaerobic conditions during preservation. The advantages of cold preservation are underpinned by the principle of decelerating metabolism during periods when the body is devoid of oxygen and nutrients. This is chiefly attributable to the fact that a mere 10 °C reduction in temperature results in a twofold reduction in most enzymatic reactions. However, studies have illuminated that even when metabolic activity decreases from 0 °C to 4 °C, the liver still necessitates oxygen for metabolic processes [33].

### 3.2. Organ Preservation Solutions for Preventing SCSI

Advancements in hepatic preservation, aimed at safeguarding allografts from IRI, have centered on the development of static cold preservation solutions boasting diverse compositions. These preservation solutions can be categorized based on their potassium levels, the type of oncotic agents employed, the presence of impermeant substances, and specific features tailored to mitigate IRI [34]. Variations in the composition of the most utilized solutions are delineated in Table 1.

The formulation and additional components of these solutions are meticulously designed to avert issues that may arise during the cooling process. As per the criteria set forth by Belzer and Southard [30], the optimal attributes of these solutions encompass (1) the prevention of cell swelling and interstitial edema induced by hypothermia, (2) the mitigation of intracellular acidosis, (3) the prevention of electrolyte imbalances, (4) the reduction of oxidative damage, especially during reperfusion, and (5) the provision of substrates relevant to cellular energy metabolism. In recent times, preservation solutions have also found utility in dynamic preservation, with their composition encompassing many of the fundamental elements utilized in static cold storage. Notably, the Belzer Machine Perfusion Solution (Belzer-MPS) serves as the primary solution employed in Hypothermic Machine Perfusion for hepatic and renal applications within the clinical domain [35,36,37].

## 4. Supercooling Strategies for Organ Preservation

Presently, SCS at +4 °C represents the benchmark for liver preservation before transplantation. This approach hinges on the principle that lower temperatures lead to a reduction in metabolic activity, thereby extending the duration of ex vivo viability. SCS at +4 °C can uphold liver viability for up to 12 h, although this timeframe may prove inadequate for certain transplantation scenarios or for long-distance transportation. Lowering the temperature further could potentially prolong storage time; however, this endeavor is hampered by the fact that tissues freeze at temperatures below 0 °C.

The University of Wisconsin (UW) preservation solution has garnered acclaim in the field of liver transplantation. Nonetheless, to enable supercooling, enriching the solution becomes imperative to mitigate cold-induced membrane damage.

Numerous publications detail the preservation of whole human livers at subfreezing temperatures without ice formation. A cryoprotective perfusion cocktail primarily comprising PEG-35 (polyethylene), 3-OMG (a glucose molecule), trehalose, and glycerol plays a pivotal role in this endeavor. The addition of 5% PEG 35 kDa to the storage medium prevents cold-induced lipid peroxidation, preserving hepatocyte viability and functionality during storage. MP is employed to alleviate the consequences of prolonged ischemia, which has been demonstrated to mitigate hypothermic endothelial damage, reactivate metabolic processes, replenish ATP levels, and mechanically prepare the vasculature for reperfusion [38,39,40,41].

When it comes to individual cells, they are often preserved in UW solution at temperatures ranging from 0 to 4 °C, akin to the conditions employed for whole organ preservation. Research has indicated that it is possible to extend the viable storage period of rat hepatocytes by incorporating 5% 35 kDa PEG into the storage medium and reducing the typical storage temperature from 4 °C to −4.4 °C, all while preventing ice formation [42]. These findings suggest that the combination of PEG supplementation and supercooling may offer an enhanced approach for the preservation of both cells and organs.

## 5. Pathophysiological Mechanisms of IRI

### 5.1. Pathophysiology of the Ischemic Cascade

While organ cooling reduces the metabolic demands placed on the tissue, certain metabolic processes persist even as the temperature decreases. During cold ischemia, hepatocytes undergo a transition from self-limiting aerobic metabolism to anaerobic metabolism, which leads to inadequate ATP production. In this context, anaerobic glycolysis becomes the sole process capable of generating ATP. Nevertheless, the rate of ATP consumption exceeds its production. Additionally, ischemia prompts the conversion of xanthine hydrogenase into xanthine oxidase, which catalyzes the degradation of hypoxanthine under aerobic conditions [43]. This enzymatic activity gives rise to superoxide radicals, a subset of ROS. The accumulation of specific metabolites, such as succinate, during the ischemic phase, has been associated with mitochondrial dysfunction in various tissues. When the organ is cooled, this metabolic reaction proceeds at a sluggish pace due to the absence of oxygen. However, upon reoxygenation of the organelle, this activity leads to an elevated generation of ROS. ROS are highly toxic as they can oxidize organic molecules and inflict damage on cell membranes through lipid peroxidation [6,9,12] (Figure 2).

#### 5.1.1. Hypothermia-Induced Cell Swelling

The temperature during cold storage impacts enzyme activity, and the inadequate ATP supply during cold ischemia leads to a near shutdown of Na^+^, K^+^-ATPase function. Consequently, the pump’s failure leads to the establishment of a sodium and potassium osmotic imbalance between the intracellular and extracellular compartments, allowing sodium to infiltrate the cell. Furthermore, intracellular proteins with negative charges (anions) amplify the influx of sodium ions. The resultant intracellular hyperosmolarity eventually triggers water uptake, culminating in cellular swelling, the formation of bulging sacs, and fragmentation. In hypothermic conditions, the membrane-bound Na^+^, K^+^-ATPase pump, responsible for maintaining osmotic equilibrium between the extracellular and intracellular spaces, becomes disrupted, further exacerbating cell swelling [43].

#### 5.1.2. Sinusoidal Endothelial Cell Injury

The primary mechanism underlying damage during cold storage involves injury to the sinusoidal endothelial cells (SECs). These cells play a pivotal role in maintaining vascular homeostasis and immune function, making them a central component of cold ischemia, whereas warm ischemia predominantly affects hepatocytes. The transcription factor Krüppel-like factor 2 (KLF2), typically expressed by liver sinusoidal endothelial cells (LSECs), is involved in vasodilation, and exerts anti-thrombotic and anti-inflammatory effects. However, during cold ischemia, KLF2, along with endothelial nitric oxide synthase, thrombomodulin, and nuclear factor erythroid 2-related factor 2, experiences down-regulation. In rat livers subjected to cold storage, the restoration of these genes helps prevent liver damage [44]. Furthermore, SECs release damage-associated molecular patterns (DAMPs) that contribute to the initiation of an immune response and express adhesive molecules that facilitate neutrophil binding [45].

Additional effects observed during cold ischemia include the development of intracellular acidosis and elevated calcium levels within the cells. Intracellular acidosis primarily results from anaerobic glycolysis, as lactate represents the final metabolic pathway byproduct. The accumulation of intracellular protons inhibits crucial glycolytic enzymes such as phosphofructokinase and further depletes ATP, thereby impeding the remaining anaerobic energy production. Additionally, intracellular acidosis activates lipoprotein lipase and lysosomal hydrolases, culminating in membrane damage and heightened permeability. Elevated intracellular calcium levels serve as a signaling factor for numerous ischemic mechanisms that ultimately lead to cell death [45].

#### 5.1.3. The Role of Mitochondria in IRI

Mitochondria, double-membrane structures located within the cytoplasm of eukaryotic cells, play multifaceted roles in ensuring cell survival and adaptation. These roles encompass the production of ATP, which provides energy for fundamental cellular processes through oxidative phosphorylation. Additionally, mitochondria are involved in numerous essential functions, including lipid, amino acid, and nucleotide metabolism, ROS signaling, calcium homeostasis, apoptosis, and safeguarding cells against oxidative stress [46]. Moreover, mitochondria have evolved signaling functions to communicate with other cells in response to environmental threats. Notably, biological responses are not initiated until mitochondrial regulatory input is integrated. For instance, mitochondrial capacity influences the selection of metabolic processes based on the cell’s energy requirements. In the physiological response to hypoxia, mitochondria serve as oxygen sensors. The production of mitochondrial ROS activates the transcription nuclear factor erythroid 2-related factor 2 (NRF2), leading to the expression of antioxidant and protective proteins. Therefore, the release of mitochondrial second messengers elicits protective antioxidant responses [47].

Dysfunctional mitochondria release components that trigger stress responses, activating various mitochondrial quality control mechanisms. These mechanisms encompass mitochondrial biogenesis, mitochondrial dynamics, and mitophagy, all aimed at optimizing mitochondrial function and preserving cellular homeostasis [48].

During the ischemic period, when nutrients and oxygen are scarce, cellular respiration is disrupted, affecting ATP synthesis. Initially, there is a reduction in ATP production, prompting the initiation of anaerobic respiration and lactate accumulation. This results in a loss of energy (ATP) and the buildup of succinat and dihydronicotinamide adenine dinucleotide (NADH). Succinate accumulation during ischemia/hypoxia becomes crucial for modulating reperfusion injury [49]. It is important to highlight that the graft responds to a lack of oxygen during cold ischemia via activating AMP kinases to avoid the intense reduction in ATP and to prevent graft injury during cold preservation [50].

Furthermore, hydrogen ions (H^+^) accumulate, leading to acidosis. Elevated intracellular H^+^ concentrations disrupt the sodium–hydrogen (Na^+^/H^+^) exchanger, causing intracellular sodium (Na^+^) accumulation. Additionally, reduced activity of the sodium–potassium adenosine triphosphatase (Na^+^/K^+^ATPase) pump exacerbates Na^+^ buildup, resulting in cell swelling and death. Widespread ischemia also disrupts sodium–calcium (Na^+^/Ca^++^) exchanger homeostasis, leading to elevated intracellular calcium levels. The increased levels of intracellular Na^+^, calcium (Ca^+^), and H^+^ ions contribute to mitochondrial deterioration, leading to the formation of mitochondrial permeability pores. Followed by concurrent apoptosis and necrosis, causing extensive damage [9].

Paradoxically, reoxygenation exacerbates mitochondrial dysfunction. The rapid restoration of efficient electron flow and subsequent ATP recovery come at the cost of reverse electron transfer (RET), an uncontrolled electron flow that generates ROS at mitochondrial complex-I. Furthermore, decreased activity of endogenous antioxidants like heme oxygenase 1 and superoxide dismutase (SOD) contributes to increased ROS production. Elevated ROS levels and mitochondrial membrane permeability both contribute to mitochondrial dysfunction and are significant contributors to hepatocyte death and IRI [12,51,52].

Mitochondria are pivotal components of all cells, such as the mitochondrial aldehyde dehydrogenase 2 (ALDH2) enzyme. ALDH2 plays a vital role in balancing damage and restoring the ability to protect cells from stress induced by hypoxia, and contributes to modulating lipoperoxidation and the generation of toxic species such as 4-hydroxy-2-nonenal (4-HNE) in IRI [53].

Mitochondrial uncoupling protein 2 (UCP2), a member of the mitochondrial anion carrier family, plays an important role in IRI, regulating processes such as cellular homeostasis, oxidative stress, and cell survival. UCP2 can serve as an oxidative stress sensor, though it is not directly involved in antioxidant defense. Additionally, UCP2 stimulates mitophagy caused by IRI, and UCP2 cardioprotective benefits in IRI are eliminated by inhibiting mitophagy [54].

Finally, the interaction of all these processes is essential for the survival of cells and organs during the processes of donation, preservation, and transplantation. It has been demonstrated that mitochondria have a pivotal role in these events and should therefore be a primary area of focus for future treatment approaches. This observation demonstrates that IRI is a multifaceted and multifactorial phenomenon that is not yet fully comprehended but plays a critical role in influencing the development of primary graft dysfunction.

### 5.2. Rewarming Injury

Following the removal of the liver and the placement of the allograft, the creation of vascular anastomosis typically takes between 30 and 60 min. During this interval, the liver gradually warms up, although reperfusion has not yet commenced. In a hypoxic environment, elevated tissue temperature leads to a heightened loss of ATP. After 30, 40, and 50 min, the liver’s temperature incrementally rises from approximately 0 °C to 12 °C, 17 °C, and finally 20 °C. During this phase, hepatocytes are more susceptible to warm ischemia when compared to SECs, since they require greater energy reserves at 20 °C [48].

### 5.3. Reperfusion Injury

#### 5.3.1. Events during Reperfusion

Tissue damage manifests within mere seconds to minutes immediately following reperfusion. This cascade of injury during reperfusion represents a response to the pre-reperfusion damage sustained by the allograft (Figure 2). Blood reperfusion, rather than promptly restoring normal conditions, exacerbates the situation. A pivotal event during this vulnerable phase is the excessive generation of ROS, including superoxide anions, hydrogen peroxide, and hydroxyl radicals. This ROS overproduction instigates abnormal cell signaling, inflicts damage on biomolecules, incites inflammation, and ultimately leads to cell death, culminating in a decline in organ function. Damage to biomolecules and ROS-mediated signaling also stimulates innate immune responses, ultimately resulting in fibrosis and the progressive deterioration of organ function. In this context, two distinct stages of reperfusion injury emerge: (1) immediate injury upon reperfusion and (2) injury accompanied by sterile inflammation, serving as an immune response to the initial injury [12,55].

#### 5.3.2. Mitochondrial Injury and Cell Death

In the immediate aftermath of reperfusion, parenchymal cells, previously injured by both cold and warm ischemia, undergo additional stress due to reoxygenation. Among the various metabolic processes, the restoration of energy production in the form of ATP is arguably the most critical determinant of cell survival during the initial stages of reperfusion. However, the damage inflicted upon mitochondria during cold ischemia and early reperfusion provides an explanation for why cells may fail to generate adequate ATP levels following reoxygenation [12]. The reintroduction of oxygen prompts a substantial surge in oxygen consumption, leading to a mitochondrial burst of ROS and reactive nitrogen species (RNS). These ROS and RNS arise from the excessive leakage of electrons from the mitochondrial electron transport chain [48,51].

#### 5.3.3. Endothelial Cell Injury Associated with Platelet and Leukocyte Adhesion

The activation of the surface of SECs plays a pivotal role in SEC injury, particularly concerning the adhesion of platelets and leukocytes during the early reperfusion phase. This activation process sets the stage for a proinflammatory environment and serves as a trigger for the inflammatory response. Upon reperfusion, LSECs express an array of cytokines along with DAMPs. Furthermore, ROS and RNS generated during reperfusion have the capacity to degrade proteoglycans, thus posing a threat to the glycocalyx. Disruption of the glycocalyx leads to the overexposure and activation of the endothelial cell surface, which, in turn, affects the adhesion of platelets and leukocytes, exerting a detrimental influence on microcirculation [52].

#### 5.3.4. Injury Caused by Sterile Inflammatory Immune Response

Ischemia/reperfusion injury is associated with an activated innate immune response involving both local and circulating immune cells [55]. The release of DAMPs initiates the activation of the first wave of resident Kupffer cells and adhesive neutrophils by binding to pattern recognition receptors on their cell surfaces [56]. One of the most prevalent pattern recognition receptors, Toll-like receptor 4 (TLR4), is present on virtually all innate immune cells. The activation of adaptive immunity can further exacerbate liver graft injury, rendering the graft more vulnerable to allograft rejection. NKT cells and T cells have been implicated as primary contributors to both ischemia/reperfusion injury and allograft rejection, with CD4+ T cells playing a central role [55]. Additionally, the complement system is activated and contributes to graft injury post liver transplantation. The deposition of the membrane attack complex, assembled by C5–9, on cellular membranes contributes to parenchymal injury. The role of the complement system in hepatic ischemia/reperfusion injury is an area of ongoing investigation [12,57].

## 6. Pharmacological Strategies for IRI Modulation

The utilization of pharmacological agents has become a significant component of organ preservation (Figure 3). Various interventions aimed at mitigating ischemia/reperfusion injury following liver transplantation have been assessed in preclinical and clinical models. These interventions encompass antioxidants, metabolism modulation, anti-adhesion strategies, anti-inflammatory agents, immunosuppression, vasodilation, and inflow modulation.

Given that hepatic ischemia/reperfusion injury involves multiple targets and mechanisms, the literature reports several therapeutic approaches. Numerous experimental studies have identified several potential natural and synthetic drugs with promise in this regard. However, the main challenge lies in identifying the most effective drugs, appropriate dosages, and optimal combinations. Despite extensive evaluations, a definitive integrative therapy for hepatic IRI injury has yet to be established.

### 6.1. Calcium Channel Blockers

The initiation of cell death is initiated by mitochondrial permeability transition pores (MPTPs), which allow the leakage of cytochrome c from the inner mitochondrial membrane, subsequently activating caspases and inducing apoptosis. This phenomenon is triggered by an elevation in intracellular calcium levels. Increased cytosolic Ca^++^ levels activate Ca^++^ uniporters in the mitochondrial membrane, leading to an accumulation of Ca^++^ within the mitochondria. When Ca^++^ homeostasis is disrupted, MPTPs form, resulting in cell injury through either apoptosis or necrosis.

Studies have indicated that pretreatment with calcium channel blockers like amlodipine can reduce the likelihood of injury by restoring balance to the cellular environment and preventing mitochondrial disturbances caused by hepatic ischemia/reperfusion [58].

### 6.2. Hormones

Corticosteroids have been employed to mitigate acute inflammation and reduce postoperative complications resulting from IRI. Their protective effects have been observed in myocardial IRI by attenuating the inflammatory response after cardiopulmonary bypass and have shown benefits in liver trauma. However, the precise role of corticosteroids in hepatic IRI remains less clear.

In a study utilizing a rodent model, groups treated with intravenous methylprednisolone before inducing ischemia demonstrated a significant reduction in their inflammatory responses. Histological analysis, serum interleukin-6 (IL-6) levels, and aspartate aminotransferase (AST) release indicated that methylprednisolone treatment provided substantial protection in normal livers compared to ischemic controls [59].

A systematic review and meta-analysis investigating the effects of perioperative steroids on IRI revealed that perioperative steroid administration is associated with favorable postoperative outcomes following hepatectomy. The treated group exhibited a significant increase in early postoperative serum interleukin-10 (IL-10) levels, a decrease in blood bilirubin levels, and reduced inflammation markers postoperatively (IL-6 and C-reactive protein). Moreover, patients receiving intravenous glucocorticoids had a 24% lower postoperative morbidity compared to the control group, and there was no evidence to support a higher risk of infectious complications or impaired wound healing [60].

Additionally, corticosteroids have been included as a component of the perfusate in MP particularly during extended perfusion, to promote graft survival [61].

Thyroid hormones play an important role in intracellular calcium homeostasis and mitochondrial function. There is extensive evidence of a sharp decrease in serum triiodothyronine (T3) levels following brain death (BD). Reduced thyroid hormone concentrations following BD have been proposed to induce hemodynamic instability, leading to a decrease in myocardial energy storage and a shift in metabolism from aerobic to anaerobic. After administration of T3 hormone to DBD, creatine phosphate and adenosine phosphate levels returned to normal levels, lactate decreased, and glycogen levels improved [62]. In a rodent model, administration of T3 prior to the induction of BD has been shown to reduce apoptosis and injury to liver cells [63]. In a retrospective study that involved 66,629 donors, it was found that T3/T4 therapy increased the number of organs that could be transplanted, but it did not have any negative effect on graft survival after transplantation [64].

### 6.3. Antioxidants

Melatonin, an endogenous antioxidant synthesized in the pineal gland, exhibits significant potential in mitigating liver IRI. Melatonin’s multifaceted role encompasses the upregulation of critical antioxidant enzymes, including super-oxide dismutase, glutathione peroxidase, and glutathione reductase, thereby enhancing cellular defense mechanisms. Moreover, melatonin has demonstrated the capacity to dampen key inflammatory pathways, such as the Toll-like receptor 4 (TLR-4)–nuclear factor kappa B (NFκB)–NOD-like receptor protein 3 (NLRP-3) axis, in murine liver models of IRI [65]. Recent murine studies have further illuminated melatonin’s role in downregulating the activity of the NFκB signaling pathway during both the early and late phases of hepatic IRI. This attenuation of NFκB activity not only ameliorates the inflammatory response but also safeguards liver function, ultimately culminating in enhanced perioperative survival rates [66]. Melatonin has also been used as additive in organ preservation solutions [67].

Ferritin, with its inherent antioxidant properties, contributes to the reduction of apoptotic stimuli in the context of IRI [68]. Mangafodipir trisodium, when administered to the donor prior to organ harvesting, has also demonstrated its efficacy in preventing IRI [69].

In the realm of natural antioxidants, ubiquinol, also known as coenzyme Q10, plays a pivotal role as a component of the respiratory chain [70]. Notably, MitoQ, a specialized formulation of ubiquinol, leverages its antioxidant potential to mitigate IRI. MitoQ’s mechanism of action involves cycling between the ubiquinol and ubiquinone forms, facilitated by complex-II, thereby perpetuating its antioxidant effects, and intervening in the IRI cascade [71].

Furthermore, N-acetyl cysteine (NAC), a precursor to glutathione, emerges as a valuable agent in protecting the liver against IRI in animal models [72]. It is used as additive in organ preservation solutions (Custodiol). Through the process of replenishing glutathione reserves and enhancing cellular homeostasis, NAC promotes overall survival and reduces the likelihood of graft dysfunction. The protective mechanisms of this additive against liver IRI have been suggested to involve the regulation of the ROS/JNK/Bcl-2 pathway, leading to the attenuation of IRI-induced autophagy and apoptosis [73].

These multifaceted interventions collectively contribute to the preservation of liver function in the face of IRI.

### 6.4. Polyethylene Glycol

Polyethylene glycol has demonstrated its ability to protect cellular structures, especially the cytoskeleton and mitochondria. It also exhibits antioxidant properties that help prevent edema-induced membrane destabilization by inhibiting cellular reactions. PEG35 also enhances nitric oxide (NO) generation, which is important in vasodilation. NO generation improves microcirculation within the graft, reducing reperfusion damage, especially interesting for fatty livers [74]. Furthermore, certain PEG molecules could cross cell membranes and prevent the induction of MPT. As a result, cell inflammation is reduced, and the membrane potential is preserved. This preservation occurs through the direct inhibition of cytochrome c release and the prevention of apoptosis [75]. PEG, at a concentration of 1 g/L, is already present in the Institute-George-Lopez-1 (IGL-1) protective solution used in SCS [76]. Institute-George-Lopez-2 (IGL-2), a modified version of IGL-1 with a PEG35 concentration of 5 g/L, was developed to be suitable for SCS and dynamic preservation, such as hypothermic oxygenated perfusion (HOPE) [77].

This new preservation solution has a higher antioxidant capacity compared to Belzer-MPS, the perfusion solution normally used for HOPE, due to its higher concentration of glutathione. Additionally, IGL2 has a lower viscosity than Belzer-MPS, facilitating graft rinsing and HOPE procedures [77]. This lower viscosity also aids in glycocalyx preservation, which enhances graft microcirculation, as well as vasodilation via nitric oxide generation [78]. The use of IGL-2 as the unique perfusion solution for static and dynamic preservation strategies would simplify the logistics of using different perfusates.

### 6.5. Other Agents

Nilotinib, an oral receptor tyrosine kinase inhibitor, has demonstrated substantial in vitro activity against c-Jun N-terminal kinase (JNK) and p38 mitogen-activated protein kinase (MAPK), both of which play pivotal roles in mediating liver IRI by modulating the expression of inflammatory factors and cell death processes. In a murine model of liver IRI, experimental studies have highlighted the noteworthy impact of nilotinib. It has been shown to effectively reduce the recruitment of inflammatory monocytes, lower the expression of pro-inflammatory cytokines, and ameliorate hepatocellular apoptosis [79].

Iloprost, an analog of prostacyclin, underwent testing in a rat model of IRI, where it displayed a reduction in hepatic enzyme levels and an upregulation in the expression of antioxidant enzymes. Notably, it also maintained the regular sinusoidal structures with normal morphology, devoid of congestion, in the IRI-afflicted rats that had been pretreated with iloprost [80]. Similarly, treprostinil, another prostacyclin analog, showcased analogous findings in a rat orthotopic liver transplantation model [81].

Heme oxygenase-1 (HO-1), a pivotal enzyme responsible for the degradation of heme molecules into biliverdin, carbon monoxide (CO), and ferrous ions, has emerged as a positive modulator of hepatic IRI prophylaxis. Biliverdin’s action in neutralizing ROS alleviates inflammatory processes, while CO, operating through the MAPK pathway, fosters an anti-inflammatory and anti-apoptotic environment crucial for maintaining microcirculation integrity. Furthermore, the regulatory influence of HO-1 on autophagy has been demonstrated to be dependent on the enzyme silent information regulator factor 2-related enzyme 1 (SIRT1). Experimental interventions involving adenovirus-mediated HO-1 gene transfer in mouse liver grafts have underscored the interplay between HO-1 and SIRT1 in controlling autophagy, offering novel insights into its multifaceted impact [82,83].

Fibrates, a class of anti-inflammatory drugs that act as direct activators of peroxisome proliferator-activated receptor alpha (PPARα), have exhibited efficacy in mitigating IRI in rat kidney models. Moreover, recent revelations have brought to light the antioxidant potential of PPARγ agonist pioglitazone, further solidifying the role of fibrates in combating IRI [84].

## 7. Dynamic Preservation

The landscape of organ preservation techniques in the field of transplantation has experienced significant evolution since the introduction of the University of Wisconsin solution in the early 1990s [85]. However, the conventional SCS method, which involves maintaining donor organs at low temperatures without continuous perfusion, has remained relatively unchanged. The limitations of SCS are apparent in its reliance on metabolic processes, particularly ATP depletion, the accumulation of metabolites, and the gradual loss of cellular membrane functions. These constraints impose a strict time limit on organ preservation using SCS.

Over the years, the criteria for organ transplantation have expanded to encompass a wider range of transplant indications. Many of the initial contraindications have been systematically addressed and overcome. This growth in transplant indications, coupled with a growing aging population with higher rates of obesity, has intensified the challenge of procuring an adequate supply of viable donor organs. Consequently, there has been a notable increase in the discard rate of donor livers, underscoring the pressing issue of organ scarcity in transplantation [1,2,5,13,17].

To address the scarcity of viable organs, efforts have been directed toward broadening the selection criteria for eligible organ donors. It is especially important to highlight that a graft will be more vulnerable to IRI the more extended criteria characteristics it has (such as steatosis, advanced age, and ischemia duration).

Additionally, ongoing investigations are focused on optimizing organ preservation techniques and devising strategies to salvage organs that were once considered marginal or unsuitable for transplantation.

A particularly promising area of innovation in organ preservation is the concept of dynamic perfusion, which has garnered renewed interest and has been the subject of numerous clinical studies [37,86,87,88,89,90]. While these technologies share the overarching goal of mitigating liver ischemia, they possess distinctive characteristics concerning the timing of their application, temperature settings, composition of the perfusate, and the types of perfusion devices employed.

Broadly, different strategies have emerged within the realm of dynamic perfusion: (1) Normothermic Regional Perfusion (NRP), (2) ex situ Normothermic Machine Perfusion (NMP), (3) Hypothermic Machine Perfusion (HMP) and (4) sequential HOPE-NMP. These approaches represent diverse methods for maintaining and revitalizing donor organs, ultimately enhancing their viability and potential for successful transplantation.

### 7.1. In Situ Normothermic Regional Perfusion

The initial approach, known as NRP, was pioneered in Spain within the context of unexpected cardiac arrest situations. In this innovative method, individuals who had the potential to become organ donors were initiated on extra corporeal membrane oxygenation (ECMO) through femoral vessel cannulation while cardiopulmonary resuscitation (CPR) was still in progress [91,92]. This pioneering technique was subsequently expanded to encompass DCD donors.

The core principle of in situ perfusion during NRP is to curtail the ischemic damage incurred by the donor organ during the warm ischemia period. It facilitates the recovery of organs before subjecting them to the cold organ flush during the organ procurement process. An additional advantage of NRP is the opportunity it offers for conducting a range of biochemical assays during perfusion. These assays can include measurements of lactate levels, transaminase activity, and pH levels, all of which enable the early assessment of the extent of IRI. This early assessment is pivotal in determining the viability of the organ.

Several comprehensive studies investigating NRP have reported superior outcomes and reduced complications, particularly in the context of DCD liver transplantation across various European countries. Building upon these favorable results, NRP has become an established standard of care, if not mandatory, for DCD donors in certain countries [91,93,94,95]. This approach represents a significant advancement in organ preservation techniques, offering enhanced outcomes and minimizing complications, including the occurrence of biliary complications.

Despite the advantage of maintaining the organ under more physiological conditions, its potential in the long-term preservation or treatment of ECD organs has not been explored. This can be explained as being mainly due to ethical and resource concerns. Furthermore, perfusion of the remaining abdominal organs implies the use of higher treatment doses than those used in the case of isolated organs. Research for specific biomarkers of organs or cell types would be necessary because multiple organs would be involved in the perfusion and interpreting the results would be challenging.

### 7.2. Ex Situ Normothermic Machine Perfusion

The second method, NMP, demonstrates the capability to extend the duration of successful organ preservation compared to conventional cold storage techniques. NMP strategies are designed to maintain the liver in a state that closely mimics physiological metabolism and synthetic liver function. This is achieved by providing the liver with oxygen and essential nutrients at a temperature of 37 °C through the utilization of a blood-based perfusate, thereby preserving physiological metabolism. When NMP is employed, the severity of hepatic IRI appears to be notably reduced [86].

NMP plays a pivotal role in maintaining the health of the liver’s endothelium and replenishing hepatic ATP reserves. Furthermore, NMP induces alterations in the expression of genes involved in liver regeneration and the regulation of inflammation. In clinical practices, the assessment of graft viability during NMP relies on various parameters, including hemodynamic stability according to the measurement of arterial and portal venous flows, resistances and pressures, bile production, perfusate lactate clearance, and the biochemical composition of the perfusate [86,87,96,97,98]. However, the quest for the most precise biomarkers for evaluating a liver’s transplantability remains ongoing [86,88,96,99]. The early initiation of NMP offers advantages since it mitigates the need for SCS, limiting IRI. Positive results have been reported in the literature regarding the successful implementation of “ischemia-free liver transplantation” (IFLT), which involves the complete avoidance of SCS [100]. However, this approach requires the transportation of NMP devices to the donation center, which complicates logistics and increases the costs of the procedure. On the other hand, the implementation of end ischemic NMP, while associated with simpler logistics and lower costs, may be somewhat less effective in preventing injuries related to ischemia [86,96,97].

NMP research initially focused on improving allograft selection and evaluation using graft viability testing, and, more recently, on rescuing rejected organs for transplantation [96], which will help ameliorate the shortage of livers for transplantation. New studies are focusing on prolonging the liver graft preservation time, which may provide more possibilities for liver transplantation, improve liver regeneration, and allow IRI modulation. However, long-term perfusions also bring new challenges, such as hemolysis, controlling oxygenation and glucose levels, removing waste products, and simulating diaphragm movements to prevent pressure necrosis. Clavien et al. [101] reported the longest preservation of a human liver, with subsequent transplantation after 68 h of NMP. Special devices are being developed to maintain perfusion for longer periods given that commercial devices only offer perfusions for up to 24 h [101,102].

### 7.3. Ex Situ Hypothermic Machine Perfusion

Perfusion strategies such as hypothermic oxygenated machine perfusion (HMP) have made significant inroads in clinical practices, particularly through techniques like hypothermic oxygenated perfusion via the portal vein (HOPE) or dual perfusion involving the portal vein and hepatic artery, known as dual HOPE (DHOPE). These methods have demonstrated considerable promise in mitigating hepatic IRI. The key to their effectiveness lies in oxygenating the perfusate during the perfusion. HMP has displayed the ability to forestall damage to mitochondria and the nucleus, as well as the activation of Kupffer cells and endothelial cells [103].

The rationale behind the success of HMP hinges on the restoration of aerobic metabolism. During ischemia, hypoxic conditions prevail within mitochondria, resulting in energy depletion and the accumulation of toxic metabolites. The reintroduction of oxygen, however, has contrasting effects. When oxygen is administered in normothermic conditions, it can trigger the production of ROS, which are the primary inflammatory mediators associated with IRI. Conversely, the reintroduction of oxygen under hypothermic conditions triggers various protective pathways within mitochondria and throughout the organs [104]. In the cold, oxygenated environment, aerobic metabolism is swiftly reinstated in mitochondria. This not only reduces the levels of detrimental metabolites like succinate and NADH, but also reestablishes ATP levels within a mere two hours. Consequently, HMP has been found to significantly mitigate IRI, inflammation, and post-transplant complications in various organs [88,90,105,106,107].

Compared to NMP, the perfusion approach employed in HMP is more cost-effective and less time-consuming. It employs a perfusate like that used in SCS. However, one of the primary challenges of HMP is the limited data available for assessing liver function. Efforts are underway to enhance the evaluation of allograft viability during HMP before transplantation. Among the most extensively studied viability markers is flavin mononucleotide (FMN). Under physiological conditions, FMN is closely associated with complex I of the electron transport chain. When the electron chain is disrupted, FMN dissociates from complex I and becomes readily detectable in acellular machine perfusates. The fluorescent properties of FMN permit the real-time measurement of this molecule in perfusates, and its levels correlate with liver function, damage, recipient complications, and graft survival following transplantation. Other parameters under evaluation include glucose, lactate, and nicotinamide adenine dinucleotide (NAD) [108,109].

### 7.4. Ex Situ Combined Hypothermic and Normothermic Dynamic Perfusion

A recently developed method involves sequential HOPE-controlled oxygenated rewarming NMP (COR), which induces a synergistic protective effect to attenuate the IRI cascade. There is increasing evidence that an abrupt temperature shift from hypothermic to normothermic perfusion may induce mitochondrial dysfunction [110,111].

In a prospective observational cohort study of 54 discarded/high-risk livers that underwent DHOPE-NMP, after functional assessment during the NMP, 34 livers (63% utilization) met the viability criteria and were transplanted. The one-year graft and patient survival rates were 94% and 100%, respectively. Post-transplant cholangiopathy occurred in one patient (3%) [111].

In this context, it is probably advisable to consider a COR strategy as an approach to rescue and assess the viability of donor livers initially rejected for transplantation or when NMP is used after a long period of static cold storage (“back-to-base” or end-ischemic).

## 8. Therapeutics Agents during Ex Vivo Machine Perfusion

Perfusion machines can be used to restore injured organs by adding therapeutic agents to the perfusate. Various treatments such as defatting agents, vasodilators, anti-inflammatory medications, gene therapy agents, and mesenchymal stem cells have been investigated in animal and human models (Table 2).

### 8.1. Defatting Cocktail

Steatotic livers, known for their heightened susceptibility to IRI, pose a substantial risk of postoperative morbidity and mortality following liver transplantation [123]. In both experimental and preclinical models, researchers have explored the application of perfusion devices to implement metabolic preconditioning of liver grafts prior to transplantation. Notably, independent studies have revealed that NMP in isolation can yield a significant reduction in steatosis in porcine models [124].

This innovative approach, which involves a “deffating cocktail (DC)”, was more recently introduced by Nagrath et al. [119] and subsequently validated in the context of mechanical perfusion in rat livers [120,122].

The DC comprises five key components: (1) PPARδ ligand GW501516 (GW5)—stimulating the transcription of genes involved in lipid oxidation and export. (2) Pregnane X receptor (PXR) ligand hypericin (HSC)—enhancing the transcription of cytochrome P450 (CYP)3A4 ammonooxygenase, promoting pro-β-oxidation. (3) Constitutive androstane receptor (CAR) ligand scoparone (SCO)—promoting the transcription of enzymes related to β-oxidation, including carnitine palmitoyltransferase. (4) Glucagon mimetic and cAMP activator forskolin (FOR)—instigating cyclic cAMP-driven β-oxidation and ketogenesis. (5) Insulin mimetic adipokine visfatin (VIS)—reducing triglyceride levels within the liver [119].

These promising findings have been corroborated in a study which involved ten discarded livers subjected to NMP. In this study, five livers were additionally treated with Nagrath’s deffating therapy, while five served as controls. The livers treated with the DC exhibited a substantial 38% reduction in their tissue triglyceride levels compared to a mere 7% decrease in the control group over a 6 h period, despite an increase in perfusate TG levels. This treatment was associated with increased tissue ketogenesis and ATP levels, as well as decreased levels of pro-inflammatory cytokines such as TNFα and IL1β, along with reduced lactate levels. Furthermore, the treated livers displayed a remarkable 40% reduction in macrovesicular steatosis upon histological examination [121].

### 8.2. Vasodilators

Effective microcirculatory support is paramount during ex vivo perfusion, particularly when dealing with severely injured livers. Microcirculation plays a pivotal role in exacerbating the damage induced by IRI and can significantly impact post-transplant outcomes. Liver sinusoidal endothelial cells are notably more susceptible to ischemic insults compared to hepatic parenchymal cells. This heightened vulnerability disrupts the normal barrier function, vascular tone regulation, and expression of adhesion molecules. Upon reperfusion, the decreased nitric oxide levels result in impaired vasodilation control. Moreover, the activation of inflammatory and coagulation cascades leads to capillary occlusion, further worsening the IRI [125].

To address these issues, vasodilators have been investigated to enhance microcirculatory integrity and subsequently improve graft viability during NMP. Among these vasodilators, prostacyclin (PGI2) exhibits the remarkable capacity to inhibit platelet aggregation, dampen leukocyte activation and chemotaxis, and curtail superoxide anion production. These actions collectively contribute to an anti-inflammatory effect while safeguarding endothelial cells from damage [118,126]. Its use has enabled successful transplantations of DCD porcine livers following NMP [117].

Another powerful vasodilator with additional antiplatelet and fibrinolytic properties is prostaglandin E1 (PGE1). Researchers have evaluated its potential to enhance microcirculation in various studies, offering a promising avenue for further investigation [115,116].

### 8.3. Other Therapeutics Agents

Incorporating anti-inflammatory agents into the perfusion process has emerged as a promising strategy [112]. Notably, the δ-opioid agonist enkephalin has garnered attention for its potential benefits in mitigating inflammation [113]. Additionally, recent reports have introduced a novel approach involving the selective NLRP3 inflammasome inhibitor mcc950, which is added to the perfusate during HMP. In a pig liver transplantation model, this innovative method showcased improved outcomes, particularly in the context of DCD organs, when compared to control groups [114].

### 8.4. Senolytics

Senescent cells are a natural consequence of the aging process and are associated with limited tissue regeneration capabilities [127]. The development of senescence involves intricate molecular processes, including chromatin modifications, the formation of senescence-associated heterochromatin foci, and the presence of senescence-associated β-galactosidase, often accompanied by the upregulation of p21CIP1 and/or p16INK4a, which contribute to senescent DNA fragmentation [128].

One hallmark of senescent cells is their release of a substantial quantity of proinflammatory factors, collectively referred to as the senescence-associated secretory phenotype (SASP) [129]. These factors exert a profound influence on tissue homeostasis and can significantly impact the functions of neighboring cells.

Emerging evidence suggests the potential utility of senolytic agents in targeting and eliminating senescent cells [130]. These agents aim to counteract the effects of senescence and hold promise in the field of research on anti-aging. Recent investigations have unveiled various cellular targets for anti-aging interventions, including members of the BCL family, HSP90, and PI3K/AKT inhibitors [131].

Senolytics have demonstrated their potential in preclinical studies by mitigating age-related dysfunction, inflammation, and multi-organ diseases. Their application may extend to addressing age-related issues in organ transplantation, particularly when dealing with organs from older donors. Administering senolytics before IRI has shown promise in reducing key markers such as cell-free mitochondrial DNA (cf-mt-DNA), Th17, and IFN+ T-cell values, ultimately diminishing the burden of senescent cells within the recipient’s body [132].

Remarkably, this therapeutic approach has been successfully integrated into perfusion machines, offering a novel method for organ preservation [133].

### 8.5. Gene Therapy

Genetic modulation represents a promising avenue for addressing the challenges inherent in solid organ transplantation, offering the potential to expand the donor pool (Table 3). Numerous gene therapy approaches have undergone extensive research, with their feasibility validated through in vivo animal studies. In allogeneic transplantation, gene therapy holds substantial promise for addressing a range of issues, including the treatment of genetic defects, congenital metabolic disorders, and coagulation abnormalities associated with donor organs [134]. Various biochemical techniques have been developed to introduce or modify genes, allowing for the precise control of gene expression. For instance, messenger RNA-based strategies employ antisense oligonucleotides (ASOs) or antagonists, while RNA interference (RNAi) leverages RNA molecules such as micro-RNA (miRNA), short hairpin RNA (shRNA), or small interfering RNA (siRNA).

In a notable study, a mouse model was employed to investigate the role of high mobility group box 1 (HMGB-1), an early mediator of reperfusion-related damage. The administration of siRNA targeting HMGB-1 resulted in improved liver function following reperfusion [139]. Another study focused on the silencing of proapoptotic caspases 3 and 8 in a mouse model of IRI, wherein the mice treated with siRNA exhibited a significantly enhanced protective response. Specifically, 30% of animals received caspase-8 siRNA treatment, while 50% were treated with caspase-3 siRNA; remarkably, they survived beyond 30 days post the ischemic event, in stark contrast to the control group, which succumbed within 5 days [140].

Among the various miRNAs found in the liver, miR-122 stands out as the most abundant. Previous research has linked miRNAs to an array of diseases and dysfunctions [141]. Notably, in an ischemic porcine cardiogenic shock model, circulating levels of miR-122 surged by nearly 460,000-fold following cardiogenic shock, far surpassing the modest elevation observed in classic markers of hepatocellular necrosis, which increased by only about 3-fold. Importantly, miR-122 levels exhibited a significant decrease following therapeutic hypothermia [142]. Additionally, during warm hepatic IRI in rats, the level of miR-122 showed a correlational increase with serum hepatic enzymes and lactate dehydrogenase (LDH) levels [143].

In the context of organ preservation during mechanical perfusion, the application of gene modulation agents, such as ASOs and siRNA, holds significant potential due to its enhanced specificity compared to systemic gene modulation.

The first study that demonstrated the application of ASOs in suppressing the pathogenicity of hepatitis C virus (HCV) in a porcine model employed ASOs that specifically targeted miRNA-122, a crucial element for HCV replication. The inhibition of miRNA expression demonstrated a substantial reduction in HCV activity [135].

The pioneering application of siRNA during mechanical perfusion systems was also reported, revealing that the direct introduction of siRNA into the perfusion system facilitated its uptake by rat liver grafts, regardless of whether hypothermic or normothermic perfusion systems were employed [136].

This same group also employed siRNA targeting p53 to modulate apoptosis in a murine model [137].

Another recent study was conducted to examine the potential of inhibiting the apoptosis-associated gene FAS using siRNA to mitigate IRI in a rat liver transplantation model. The study resulted in inconclusive findings regarding the efficacy and feasibility of gene modulation during MP to enhance graft function. However, the study did demonstrate that absorption of liver siRNA occurs during HMP [138].

These results underscore the potential of siRNA-based approaches in the context of organ preservation and transplantation.

### 8.6. Mesenchymal Stem Cells

Mesenchymal stem cells (MSCs) have emerged as a promising avenue for improving organ transplantation outcomes (Table 4) by modulating the inflammatory response and reducing tissue damage [144]. MSCs, derived from various tissues, possess the capacity not only to replace damaged cells but also to exert beneficial effects through paracrine mechanisms [145]. These mechanisms include anti-inflammatory, antiapoptotic, antioxidant, antifibrotic, and proangiogenic properties mediated by the release of extracellular vesicles (EVs), cytokines, and growth factors [146]. The impact of MSCs and their EVs on the repair of target organs involves a range of mechanisms. These mechanisms encompass the mitigation of IRI, the stimulation of angiogenesis, the regulation of target cell proliferation, and the inhibition of the epithelial–mesenchymal transition [147]. One study underscored the potential of MSCs in enhancing graft survival rates when incorporated into the donor organ before transplantation [148].

With the advent of perfusion machines capable of maintaining organs under physiological conditions outside the human body, there is now an opportunity to directly administer MSC therapies to the perfused graft. Pioneering work by Sasajima et al. [149] has demonstrated successful reconditioning of rat livers procured after DCD by administering MSCs during NMP. Swine adipose MSCs administered at the initiation of perfusion with blood-free perfusate exhibited increased bile production and preserved sinusoidal and hepatocellular morphology after 2 h of NMP.

Another study successfully demonstrated the internalization of extracellular vesicles derived from human liver stem cells during ex situ perfusion. Following this, the livers that were treated exhibited reduced histological damage and lower levels of liver injury markers following a 4 h perfusion period [150].

Building on these initial findings, several experiments have been conducted, yielding significant advances in in vitro liver preservation techniques, and optimizing MSC culture methods.

Nevertheless, it is important to note that the application of MSC therapy in this context remains in the early stages of clinical trials and laboratory research, signifying ongoing exploration and development.

## 9. Summary

The principal contributors to organ shortages lie in the diminishing quality of donor organs within the aging population and associated pathological conditions such as obesity. Within this context, the utilization of extended criteria donor (ECD) grafts has gained prominence [5,13,17].

IRI stands out as one of the foremost challenges in the field of liver transplantation, and its severity directly influences graft function and post-transplant outcomes. This is particularly pertinent in ECD grafts, which exhibit a heightened susceptibility to IRI, exacerbated by their absence of reliable biomarkers for pre-transplantation assessments of graft function [4,6,7,12].

The cumulative impact of graft IRI throughout liver transplantation procedures, encompassing organ recovery, static and dynamic graft preservation, graft implantation, and reperfusion, plays a pivotal role in determining the success of the transplantation outcome. In this context, our efforts have concentrated on minimizing IRI and, more recently, exploring its treatment to leverage marginal livers for expanding the donor pool.

Addressing the challenge of formulating a viable therapeutic strategy for preventing and treating IRI necessitates an exhaustive understanding of the mechanical and molecular foundations of hepatic IRI. In this light, a diverse range of pharmacological therapies and technical static and dynamic preservation strategies has been scrutinized for the prevention of multifactorial IRI, as succinctly outlined in this review.

Recent years have witnessed a significant shift in abdominal organ transplantation preservation towards an increased application of dynamic preservation techniques. In this scenario, a novel preservation solution, IGL-2, emerges with distinct advantages over common perfusion solutions. This innovative preservation solution holds the promise of providing a singular perfusion solution for static and dynamic preservation strategies, streamlining the logistics of using diverse perfusates [76,77,78].

Broadly, three primary strategies in dynamic preservation are presently applied to human organs before implantation: Normothermic Regional Perfusion (NRP), ex situ Normothermic Machine Perfusion (NMP), and Hypothermic Machine Perfusion (HMP).

A promising strategy has been developed via a sequential combination of HOPE-controlled oxygenated rewarming NMP (COR) [111].

Most of the existing research studies on therapeutic interventions have been conducted within the realm of NMP. This technique leverages the treatment of a metabolically active organ to modulate or potentially reverse a disease process in the donor organ, thereby optimizing it for transplantation.

The current challenges revolve around implementing standardized protocols and guidelines into daily clinical practice. These protocols and guidelines seek to assess the risks associated with specific donors, determine utilization rates, and establish criteria for using perfusion techniques to facilitate the standardized use of machine perfusion (MP) for ECD organs [156]. Such implementation would empower surgeons to accurately assess the ECD graft, determine the most appropriate perfusion strategy, and ultimately decide on its suitability for transplantation.

## 10. Concluding Remarks

Future endeavors in liver transplantation must concentrate on comprehending the processes linked with IRI, enhancing techniques for evaluating liver quality and function pre transplantation, improving the understanding of graft viability, and refining methods for reconditioning marginal grafts before implantation in the recipient. The ultimate goal in organ transplantation is to establish perfusion protocols that minimize and modulate IRI, achieve hepatic regeneration of ECD grafts, and provide reliable biomarkers for organ viability evaluation and prediction of post-transplant outcomes. 

## Figures and Tables

**Figure 1 ijms-25-01117-f001:**
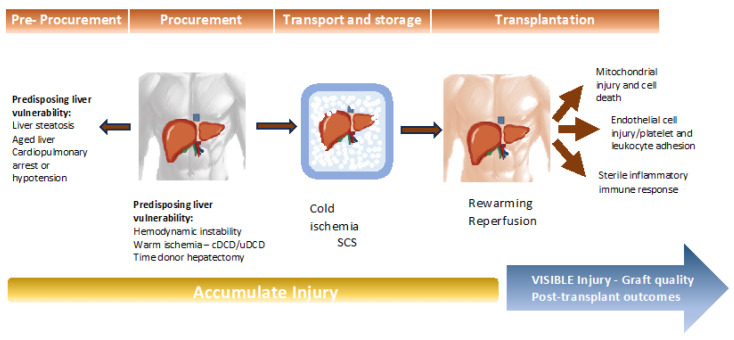
Process of injury throughout the pre-procurement, procurement, preservation, and transplantation periods.

**Figure 2 ijms-25-01117-f002:**
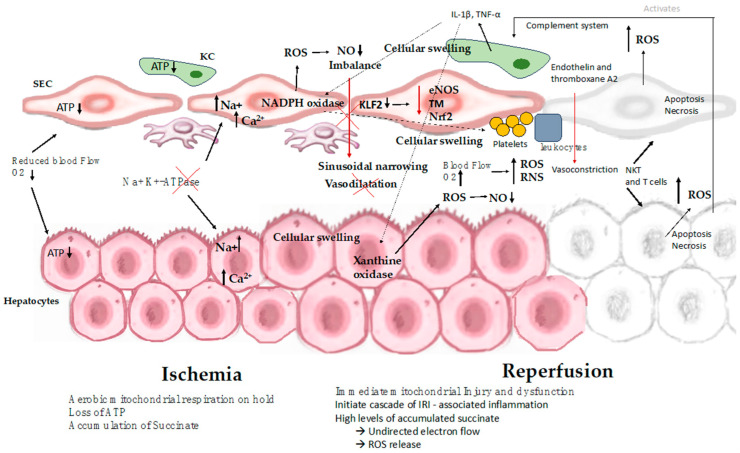
Mechanisms involved in hepatic ischemia-reperfusion injury. The process of IRI involves a series of complex events, such as mitochondrial deenergization, metabolic acidosis, ATP depletion, Kupffer cell activation, calcium overload, oxidative stress, and the upregulation of pro-inflammatory cytokine signal transduction, resulting in cellular swelling, apoptosis, and necrosis. ATP: adenosine triphosphate; SEC: sinusoidal endothelial cell; KC: Kupffer cell; IL-1β: interleukin-1β; NKT: natural killer T cell; NO: nitric oxide; ROS: reactive oxygen species; T cell: CD4+ T lymphocyte; TNF: tumor necrosis factor; eNOS: endothelial nitric oxide synthase; TM: thrombomodulin; Nrf2: nuclear factor erythroid 2-related factor 2.

**Figure 3 ijms-25-01117-f003:**
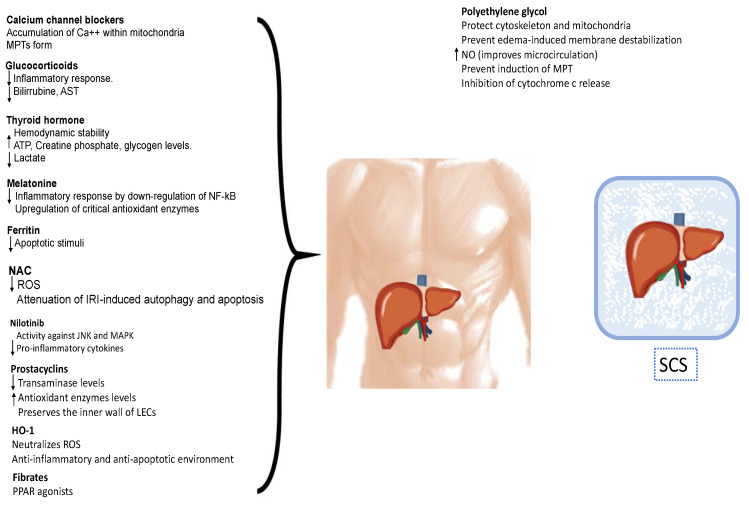
Pharmacological strategies for protect the liver against ischemia–reperfusion injury. These interventions encompass antioxidants, metabolism modulation, anti-adhesion strategies, anti-inflammatory agents, immunosuppression, vasodilation, and inflow modulation, reducing the inflammatory response, inhibiting the apoptosis of hepatocytes, and promoting the regeneration of damaged liver tissue. MPT: mitochondrial permeability transition pores; ATP: adenosine triphosphate; NF-κB: nuclear factor-κB; ROS: reactive oxygen species; IRI: ischemic reperfusion injury; JNK: c-Jun N-terminal kinase; MAPK: mitogen-activated protein kinase; LECs: sinusoidal endothelial cells; HO-1: Heme oxugenase-1; PPAR: peroxisome proliferator-activated receptor; NO: nitric oxide; MPTPs: mitochondrial permeability transition pores; SCS: static cold storage.

**Table 1 ijms-25-01117-t001:** Characteristics and compositions of current solutions for static and dynamic perfusion.

	BEL-GEN (UW)	PERF-GEN (Belzer MPS) *	CE	HTK	HTK-N	IGL-1	IGL-2
**Electrolytes (mmol/L)** Potassium	125	25	100	10	10	30	25
Sodium	30	120	15	15	16	120	125
Magnesium	5	5	13	4	8	5	5
Chloride		1		50	30	20	
Calcium			0.25	0.015	0.02		
Zinc							0.091

**Impermeant** **substances (mmol/L)**							

Lactobionic acid	100		80			100	100
Mannitol		30	60				60
Gluconate		85		30			
Ribose		5					
Raffinose	30					30	
Glucose		10			33		
**Oncotic agents (g/L)**							
HES	50	50					
PEG-35						1	5
**Buffers**	Phosphate Sulfate	Phosphate HEPES	Histidine Bicarbonate	Histidine	Histidine N-acetylhistidine	Phosphate	Phosphate
							HEPES
							Histidine
							Sulfate
**Antioxidants**	Glutathione Allopurinol	Glutathione	Glutathione	Tryptophan	Tryptophan	Glutathione Allopurinol	Glutathione
**Metabolic precursors**	Adenosine	Adenine		α-ketoglutarate	α-ketoglutarate	Adenosine	Adenosine Sodium nitrite
					Alanine		
					Arginine		
					Aspartate		
					Glycine		
**Osmolarity (mOsm/L)**	320	300	320–360	310	305	320	360
**pH**	7.4	7.4	7.3	7.2	7.0	7.4	7.4
**Viscosity (cP)**	5.7	2.40	1.15	1.8	1.8	1.28	1.7

UW: University of Wisconsin solution; CE: Celsior solution; HTK: Histidine-tryptophan-ketoglutarate solution; IGL: Georges Lopez Institute solution; MPS: machine perfusion solution; HES: hydroxyethyl starch; PEG: polyethylene glycol. * Dynamic perfusion solution.

**Table 2 ijms-25-01117-t002:** Therapeutic agents in models of ex vivo liver machine perfusion.

Author/Year	Function	Therapeutic Agents	Perfusion	Model	Time in MP	Outcomes
Goldaracena et al., 2016 [112]	Anti-inflammatory agents	- Alprostadil - N-acetilcisteine - Carbon Monoxide - Sevofluraneo	SNMP	Pig	4 h	- **Perfusion**: Lower AST levels vs. control at 1, 2, and 3 h. Reduced IL 6, TNFα, and galactosidase levels. Increased IL10 levels. - **After transplantation**: Lower bilirubin levels at 1 and 3 days. Decreased hyaluronic acid (marker of improved endothelial cell function) at 1, 3, and 5 h after reperfusion.
Beal et al., 2019 [113]	δ-opioid agonist	Enkephalin	NMP	Murine	4 h	- **Perfusion**: Protective against OE and decrease in phosphorylation of JNK and p38. Lower perfusate ALT and tissue MDA. Activation of the PI3K/Akt signaling pathway. Better tissue ATP and glutathione. Preserved tissue architecture.
Yu et al., 2019 [114]	NLRP3 inflammasome inhibitor	mcc950	HMP	Pig	2 h	- **After transplantation**: Lightest IRI. Lower MEAF and degree of hepatocytes injury. Lowest NLRP3 pathway activation. Lower TNF-α, IL-1β, β-galactosidase, ALT/AST, MDA, apoptosis staining, and caspase-1 levels.
Hara et al., 2013 [115]	Vasodilators	Prostaglandin E1	NMP	Murine	1 h	- **Perfusion**: Bile production similar in uDCD and DBD. Decreased AST/ALT and TNFα levels. Necrosis and apoptosis decreased. Ameliorated induction of MPT, and mitochondrial cytochrome C. Suppression of cytosolic JNK activation and mitochondria Bax. Decrease in mitochondrial Bcl-2 was suppressed.
Maida et al., 2016 [116]	Vasodilators	Prostaglandin E1	NMP	Murine	30 min	- **After transplantation**: Lower AST/ALT, MDA, ICAM-1, and cellular damage. Greater bile production. Higher ATP.
Nassar et al., 2014 [117]	Vasodilators	Prostacyclin (epoprostenol)	NMP	Pig	10 h	- **Perfusion**: Lower AST/ALT and LDH levels. Higher bile production. Preserved hepatic architecture
Echeverri et al., 2018 [118]	Vasodilators	BQ123, epoprostenol, and verapamil	NMP	Pig	3 h	- **Perfusion**: Higher HAF with BQ123. Lower AST with BQ123 and verapamil. Lower TNFα and IL-6 with BQ123. Lower beta galactosidase with BQ123 and verapamil. - **After transplantation**: Lower AST with BQ123 and verapamil. Lower INR, ALP, and bilirubin with BQ123 and verapamil.
Nagrath et al., 2009 [119]	Defatting cocktail	Forskolin, GW7647, scoparone, hypericin, visfatin, and GW501516.	NMP CFH	Murine	3 h	- Decreased intracellular lipid content of CFH by 35%/24 h, and perfused livers by 50%/3 h. Greater bile production and oxygen uptake rate. Increase in β-oxidation and triglyceride secretion.
Liu et al., 2013 [120]	Defatting cocktail	Forskolin, GW7647, scoparone, hypericin, visfatin, and GW501516.	SNMP	Murine	6 h	- **Perfusion**: Increase in VLDL and triglyceride content in perfusate (with or without a defatting cocktail).
Boteon et al., 2019 [121]	Defatting cocktail	Forskolin, GW7647, scoparone, hypericin, visfatin, and GW501516.	NMP	Discarded human livers	12 h	- **Perfusion**: Reduced triglycerides by 38% and macrovesicular steatosis by 40% over 6 h. Improvement of bile production and bile pH, lower vascular resistance, and ALT. Down-regulation of markers for oxidative injury and immune cells (CD14; CD11b). Reduced the release of TNFα and interleukin 1β.
Lin et al., 2021 [122]	Defatting cocktail	Forskolin, GW7647, scoparone, hypericin, visfatin, and GW501516.	HMP	Murine	3 h	- **Perfusion**: Protective effect from microcirculation disturbance and endoplasmic reticulum stress. Improve ATP and glycogen synthesis. - **After transplantation**: Prevention of nuclear injury and endothelial damage.

NMP, normothermic machine perfusion; HMP, hypotermic machine perfusion, SNMP, subnormothermic; MDA, malondialdehyde; ATP, adenosine triphosphate; MEAF, model for the early allograft function; uDCD, uncontrolled non-heart-beating model; DBD, heart-beating donor; MPT, mitochondrial permeability transition; ICAM-1, intercellular adhesion molecule 1; BQ123, endothelin1 antagonist; HAF, hepatic artery flow; CFH, cultured fatty hepatocytes; OE, oxidative stress.

**Table 3 ijms-25-01117-t003:** Gene modulation approaches during ex vivo liver machine perfusion.

Author/Year	Function	Therapeutic Agents	Infusion	Perfusion	Model	MP Time	Outcomes
Goldaracena et al., 2017 [135]	Sequesters miRNA-122 and inhibits HCV replication.	ASOs (miravirsen)	Delivered in perfusion solution.	NMP	Pig	12 h	NMP improved miravirsen uptake versus SCS. Significant miR-122 sequestration and miR-122 target gene depression. Suppression of HCV replication after established infection and prevention of HCV infection.
Gillooy et al., 2019 [136]	The Fas receptor expressed in liver signals hepatocytes to apoptosis.	siRNA (against Fas receptor)	Delivered in perfusion solution via portal vein cannulation.	NMP and HMP	Murine	4 h	siRNA added directly to perfusion solution is absorbed into rat livers during NMP and HMP.
Thijssen et al., 2017 [137]	p53 tumor suppressor, a transcription factor which can induce cell apoptosis.	siRNA (against the p53 gene)	-	NMP	Murine	6 h	siRNA is capable of reaching and penetrating liver cells. HE stains showed less vacuolization and less cell infiltration. Less positive cells in immunofluorescence for p53. Lower levels of inflammatory cytokines (IL-1, IL-6, and TNFα), neutrophil infiltration, and lipoperoxidation.
Bonaccorsi-Riani et al., 2022 [138]		siRNA (against Fas receptor)	Added to the perfusate.	HMP	Murine	1 h	Increased anti-inflammatory cytokines.

ASOs, antisense oligonucleotides; SCS, static cold storage; NMP, normothermic machine perfusion; HMP, hypothermic machine perfusion; HCV hepatitis C virus

**Table 4 ijms-25-01117-t004:** Mesenchymal stem cells during ex vivo liver machine perfusion.

Author/Year	MSC Sources	Model	Type of Machine	Time	Infusion	Outcomes
Sasajima et al., 2018 [149]	Swine adipose MSC	30 min warm ischemia murine, DCD	NMP	2 h	Injected into the portal vein	Increase bile production; improve narrowing of the sinusoidal space
Rigo et al., 2018 [150]	Human HLSC-EVs	Murine	NMP	4 h	Added to the circuit 15 min after starting perfusion	Lower AST and LDH; reduced histological damage; reduced hepatocyte apoptosis
Yang et al., 2020 [151]	Rat BMMSCs	30 min warm ischemia murine, DCD	NMP	8 h	Injected via portal vein	Improved liver function markers and liver histological damage, reduced hepatocyte apoptosis, and repaired hepatocyte mitochondrial damage.
Verstegen et al., 2020 [152]	Human BMMSCs	15–45 min warm ischemia pig, DCD	HMP NMP	0.5 h 4 h	Infused during HMP	Increased IL-6 and IL-8 Immunomodulatory effects
Laing et al., 2020 [153]	Human MAPCs	Discarded human livers	NMP	6 h	Infused directly into the right lobe via the hepatic artery or portal vein	Immunomodulatory effects
Sun et al., 2021 [154]	Rat BMMSCs	30 min warm ischemia murine, DCD	NMP	6 h	Added to the perfusate	Lower Suzuki’s score Reduced the level of ROS and free Fe^2+^ Increased bile production
De Stefano et al., 2021 [155]	Human HLSC-EVs	60 min warm ischemia murine, DCD	NMP	6 h	Added to the circuit 15 min after starting perfusion	Reduced transaminase release Enhanced liver metabolism Higher bile production (higher dose) Lower intrahepatic resistance (higher dose) Reduced necrosis and enhanced proliferation

MSCs, Mesenchymal stem cells; HLSC-EVs, human liver stem cell-derived extracellular vesicles; BMMSCs, bone marrow mesenchymal stem cells; MAPC, multi-potent adult progenitor cells.

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
