# Peer review of "Focusing on Ischemic Reperfusion Injury in the New Era of Dynamic Machine Perfusion in Liver Transplantation"

_ijms, 2024, doi:10.3390/ijms25021117_

Round 1
Reviewer 1 Report
Comments and Suggestions for Authors
1. Title- I would recommend to focus the title on the liver transplantation related IRI as the main target for the readers.
2. Methodology- please consider to mention the inclusion/exclusion criteria for data research. A PRISMA flow diagram could be desirable
3. Introduction- please add at the end of this subsection a short sentence about the aims of this review
4. Figure- please add the figure caption
5. Lines 285-300. Please provide a comprehensive figure on this topic
6. Please provide for the subsection 5.3, an explanatory figure
7. I would recommend to integrate the subsection 5.4 in the subsection 5.1, since the mitochondria plays a pivotal role in hepatic energy imbalance during IRI
8. Regarding subsection 6, I would recommend to include an introductive flow-chart/graph with the main therapeutic strategies and their role in preventing/treatment IRI
Author Response
We appreciate your prompt attention and constructive criticism of our manuscript entitled "Focusing on Ischemic Reperfusion Injury in the new era of dynamic machine perfusion" to improve the quality and depth.
We followed yours suggestions:
- Title: Focusing on Ischemic Reperfusion Injury in the new era of dynamic machine perfusion in liver transplantation
- Methology: We consider that, since we have not performed a systematic review or meta-analysis, it is not necessary to follow a predefined protocol in the literature search. Instead, it focuses on the critical interpretation of the available evidence.
- Purpose: The aim of this review is to provide an overview of the underlying mechanisms of liver IRI and address different protective strategies used during ex vivo liver machine preservation, focusing on current clinical applications and future perspectives in liver transplantation.
- Done
- Done
- Done
- Done
- Done
Should you require any further information or have additional inquiries, please do not hesitate to contact us.
Thank you very much
Reviewer 2 Report
Comments and Suggestions for Authors
This is a comprehensive review of innovative research and application of machine perfusion to minimize ischemic reperfusion injury in liver transplantation.
There are just a few minor comments
1. line 102 instead of fresh you should may be use "new" therapeutic targets
2. Coming back to basics regarding measures to protecting organ injury in the brain dead you should mention the use of steroids and thyroid hormone.
3. line 699 instead of hemodynamic stability that is not measured during machine perfusion I would use perfusion parameters (flow/pressure and resistance)
3. I suggest to add section 7.4 on the combined use of machine perfusion techniques with reference to a recent clinical study by Otto B van Leeuwen published in Am J Transplant . 2022 Jun;22(6):1658-1670. doi: 10.1111/ajt.17022. Epub 2022 Apr 18. Sequential hypothermic and normothermic machine perfusion enables safe transplantation of high-risk donor livers
4. In Table 2 there is missing information on reference specifying the results only (after Echverri et al)
Author Response
Hello
We appreciated your suggestions and constructive criticisms, which have significantly contributed to enhancing the quality and depth of our work.
- Done
- Add: Thyroid hormones play an important role in intracellular calcium homeostasis and mitochondrial function. There is extensive evidence of a sharp decrease in serum triio-dothyronine (T3) levels following brain death (BD). Reduced thyroid hormone concen-trations following BD have been proposed to induce hemodynamic instability, leading to a decrease in myocardial energy storage and a shift in metabolism from aerobic to an-aerobic. After administration of T3 hormone to DBD, creatine phosphate and adenosine phosphate levels returned to normal levels, lactate decreased, and glycogen levels im-proved. In a rodent model, administration of T3 prior to the induction of BD has been shown to reduce apoptosis and injury to liver cells. In a retrospective study that in-volved 66,629 donors, it was found that T3/T4 therapy increased the number of organs that could be transplanted, but it did not have any negative effect on graft survival after transplantation.
- Done
-
Add: 7.4 Ex situ combined hypothermic and normothermic dynamic perfusion
A recently developed method involves sequential HOPE-controlled oxygenated rewarming NMP (COR), which induces a synergistic protective effect to attenuate the IRI cascade. There is increasing evidence that an abrupt temperature shift from hypothermic to normothermic perfusion may induce mitochondrial dysfunction. In a prospective observational cohort study of 54 discarded/high risk livers that underwent DHOPE-NMP, after functional assessment during the NMP, 34 livers (63% utilization) met the viability criteria and were transplanted. One-year graft and patient survival were 94% and 100%, respectively. Post-transplant cholangiopathy occurred in 1 patient (3%). In this context, is probably advisable to consider a COR strategy as an approach to rescue and assess the viability of donor livers initially rejected for transplantation or when NMP is used after a long period of static cold storage ("back-to-base" or end-ischemic).
5. We have checked the table, and the information appears to be complete. We believe that the missing information could be due to format incompatibility between word processors software.
Thank you very much